# Burden of Malaria in Sao Tome and Principe, 1990–2019: Findings from the Global Burden of Disease Study 2019

**DOI:** 10.3390/ijerph192214817

**Published:** 2022-11-10

**Authors:** Yuxin Wang, Mingqiang Li, Wenfeng Guo, Changsheng Deng, Guanyang Zou, Jianping Song

**Affiliations:** 1Artemisinin Research Center, Guangzhou University of Chinese Medicine, Guangzhou 510440, China; 2School of Public Health and Management, Guangzhou University of Chinese Medicine, Guangzhou 511495, China

**Keywords:** malaria burden, incidence, mortality, DALY, Joinpoint regression

## Abstract

**Background**: Malaria is a parasitic infection transmitted by mosquito vectors, commonly found in tropical regions, and characterized by high morbidity and mortality. It causes a heavy disease burden in Sao Tome and Principe (STP), an island country in West Africa which at one time had a high incidence of malaria. **Objective**: This study aims to analyze the trend of disease burden of malaria in STP. **Methods**: The crude and age-standardized incidence, mortality, and disability-adjusted life years (DALYs) rate data of malaria were extracted from GBD 2019. Joinpoint 4.9 software was used to calculate the annual percentage change (APC) and the average annual percentage change (AAPC), which were also used to indicate the change in disease burden by different stages. **Results**: In general, the age-standardized incidence rate (ASIR), age-standardized mortality rate (ASMR), and age-standardized DALYs rate (ASDR) of malaria presented a decreasing trend between 1990 and 2019, with an average annual decrease of 5.6%, 6.2%, and 10.7%, respectively, in STP. Specifically, all indicators first presented an increasing trend from 1990 to about 2000, followed by a decreasing trend until 2019, although the incidence rebounded slightly after 2015. Overall, the ASIR, ASMR, and ASDR of malaria reduced by 77.08%, 87.84%, and 82.21%, respectively, in 2019 as compared to 1990. No significant differences in disease burden were found between males and females between 2005 and 2019. Children who were under 5 years old showed a relatively small decrease in the rate of DALYs as compared to other age groups, but remained the group with the highest disease burden of malaria in the country. **Conclusions**: The disease burden of malaria in STP showed a significant decrease between 1990 and 2019, but it will still be challenging to achieve the goal of eliminating malaria by 2025. The government and relevant authorities should aim to strengthen the prevention and surveillance of malaria and tailor population-specific interventions in order to reduce the disease burden of malaria in STP.

## 1. Introduction

Malaria is a parasitic disease, caused by Plasmodium and transmitted by Anopheles mosquitoes, which not only poses a serious health risk to the population but also imposes a significant disease burden on individuals, families, and the whole society. According to the World Malaria Report 2020 [1], globally, there were an estimated 229 million cases of malaria in 87 malaria-endemic countries in 2019, including approximately 215 million cases in Africa. Since 2000, the African region has seen a 44% reduction in malaria deaths and a 67% reduction in malaria mortality over the same period, making great strides in the process of malaria elimination. As part of its efforts to achieve Sustainable Development Goals (SDG) [2], the World Health Organization (WHO) launched the global technical strategy (GTS) for malaria, which aims to reduce the incidence and mortality of malaria by at least by 90% by 2030 [3]. Sub-Saharan Africa is the most affected region and accounts for a high proportion of malaria cases and deaths [4]. Reducing the burden of malaria, particularly in sub-Saharan Africa, is important to achieve the SDGs.

The Democratic Republic of Sao Tome and Principe (STP) is an island country located in the southeastern Gulf of Guinea on the west-central side of Africa. It has a tropical rainforest climate with year-round humidity and heat, and malaria can spread throughout the year, with higher incidence between November and January and between May and June. Malaria has not only caused a heavy burden on the people and society of STP in terms of health and life, but also in the terms of economic burden. In 2018, the per capita expenditure on malaria control in STP was $16, the same as in 2017 [4]. STP, like many the sub-Saharan African countries, does not own strong surveillance and health-management information systems to accurately measure the mortality and incidence of malaria [5]. Despite the low prevalence of malaria currently observed, malaria is still considered one of the most serious public health problems in STP, where the main Plasmodium species is Plasmodium falciparum [6]. To ensure access to malaria treatment and diagnosis for the population, a Ministerial Order was issued in 2008 that mandates free ACTs, as well as free public diagnosis of malaria in the public sector for all age groups [7]. According to a malaria case-management protocol developed by the STP government (revised in 2018), the country plans to eliminate local malaria cases by 2025.

Knowledge of the epidemiology of malaria is invaluable in assessing the burden of the disease [8]. Among sub-Saharan African countries, scholars in Ethiopia [5], Zimbabwe [9], and other regions have elaborated on the malaria trends in their countries by using data from GBD. However, few studies have been conducted in terms of malaria epidemic trends in STP. In addition, most studies only analyze the overall trend of the malaria burden in the study period, and do not provide a specific description of each stage in the interval based on gender and age. This paper analyzes the burden of malaria in STP from 1990 to 2019, by reporting the epidemiological trends of malaria based on gender and age using the GBD 2019 data.

## 2. Methods

### 2.1. Study Context

The islands of Sao Tome and Principe are located approximately 360 and 269 km from the west coast of the African continent, respectively, and Principe is located 160 km north of Sao Tome. The country’s territory extends over 1001 square kilometers, of which Sao Tome is 859 square kilometers and Principe is 142 square kilometers. According to the Health Statistics Yearbook of STP 2020, the National Health System is divided into four health regions, namely the Northern Regional Health Region (Lobata and Lembá Regions), the Central Regional Health Region (Água Grande and Mé-Zóchi Regions), the Southern Region (Caué and Cantagalo Regions), and the Autonomous Region of Principe Health Region (Principe Region). In 2019, there were approximately 206,426 people nationwide, with a malaria incidence rate of 3281.93 per 100,000 (3282.91 per 100,000 for males and 3280.95 per 100,000 for females), and a mortality rate of 6.03 per 100,000 (6.04 per 100,000 for males and 6.02 per 100,000 for females). There is a total of 1486 health professionals in the health system in this country. There are two hospitals in Sao Tome and Principe, one of which is the national hospital in the capital (Sao Tome), holding 72% of the hospital beds in the country, and which hosts the country’s major medical activities, while the other is a regional hospital (Autonomous Region of Principe). There is one health center in each district and there are 32 health sites throughout the country. There is a public pharmacy in each health post, with very few private pharmacies. There is a National Center for Endemic Diseases (CNE) as the main department dealing with malaria. There were 68 staff in 2019, with relatively low complete expertise and capacity.

### 2.2. Data Sources

The major sources of data for this research were obtained from the GBD 2019 public database (http://www.healthdata.org/) (accessed on 25 February 2022). The 2019 Global Burden of Disease, Injuries, and Risk Factors Study (GBD 2019) is a regional and global assessment conducted to measure the incidence and prevalence of diseases and the mortality, incidence, and disability rates due to major diseases, injuries, and risk factors worldwide [10]. GBD 2019 used a Bayesian meta-regression DisMod-MR 2.1 model to estimate and analyze the data to ensure consistency of data estimation. Each of these types of data is identified from a systematic review of published studies, searches of government and international organization websites, published reports, primary data sources such as Demographic and Health Surveys, and contributions of datasets by GBD collaborators [11]. Based on data indicators obtained from the IHME, this study reports the changes in the burden of malaria in STP from 1990 to 2019 to inform the assessment of the disease burden and the development of future malaria elimination strategies in STP. IHME analyzed the most recent estimates of the world health status for 369 diseases and injuries and 87 risk factors between 1990 and 2019. Based on the GBD 2019 data, this paper analyzes participants into five age groups (i.e., <5 years, 5–14 years, 15–49 years, 50–69 years, and ≥70 years). The cause of malaria was defined and identified according to the International Classification of Diseases, 10th (ICD-10). The ICD-10 codes for Malaria are B50-B50.0, B50.8-B52.0, B52.8-B53.1, B53.8-B54.0, and P37.3-P37.4.

### 2.3. Indicators of Disease Burden Analysis

This paper uses incidence, mortality, DALY rate, YLL rate, and YLD rate to describe the disease burden of malaria. Disability-adjusted life years (DALYs) measure the loss of health due to fatal or nonfatal disease burden, which is the sum of years of life lost due to premature death (YLLs) and the years lost due to disability (YLDs) [12]. YLLs are years lost due to premature mortality, which can be calculated by subtracting the age at death from the longest possible life expectancy for a person at that age. YLDs are an abbreviation for “years living with disability”, which can also be described as years living in less-than-ideal health.

### 2.4. Statistical Analysis

Data were input and collated using Excel 2019 to describe the distribution of STP’s malaria incidence, mortality, and DALY rate, by year, gender, and age from 1990 to 2019, with the main data described for the years 1990, 1995, 2000, 2010, 2015, and 2019. The magnitude of change in each indicator of disease burden from 1990 to 2019 was calculated by performing Joinpoint regression analysis through the use of Joinpoint 4.9.0.0 software (National Cancer Institute, Information Management Services, Inc, United States). The Joinpoint regression model was used to describe changes in the disease burden of malaria over the period from 1990 to 2019. It was fitted to the trend of change in disease burden from 1990 to 2019 for the overall population, males, females, and age groups, using time as the independent variable and standardized incidence, standardized mortality, and standardized DALY rates as the dependent variables. *Pairwise comparison* was used to perform the overlap parallelism test to determine whether the differences in trends of change were statistically significant for males, for females, and for each age group at α = 0.05. The model identifies the inflection points for trend shifts by creating segmented regressions and calculates the average annual percentage change (AAPC), the annual percentage change (APC), and their 95% confidence intervals (CI). The AAPC is the average annual trend over the entire period and the APC is the average annual trend within each period, which plays a role in quantifying the magnitude of change in the disease burden of malaria over the entire period and determining whether the trend change is statistically significant. In the absence of any join points, APC becomes constant; thus, it equals the AAPC. Otherwise, the whole period is segmented by the points with a trend change [13]. If APC < 0, it indicates a decreasing trend; by contrast, APC > 0 indicates an increasing trend, and if APC= AAPC, it indicates a monotonic change with no turning point [14].

## 3. Results

### 3.1. Descriptive Analysis of Incidence, Mortality, and DALY Rate, 1990–2019

The incidence, mortality, DALY rate, age-standardized incidence, age-standardized mortality, and age-standardized DALY rate of malaria showed an increasing and then decreasing trend from 1990 to 2019 overall. All indicators of the disease burden of malaria in 2019 decreased by more than 75% compared to 1990, with a decrease of 77.08% in ASIR, 87.84% in ASMR, and 82.21% in ASDR per 100,000 inhabitants (Table 1). Detailed analysis of the trend of age-standardized incidence between 1990 and 2019 is presented below.

The incidence of malaria in STP from 1990 to 2019 showed an increasing and then decreasing trend, with an overall decreasing trend. The overall incidence of malaria decreased from 18,089.54 per 100,000 in 1990 to 3281.93 per 100,000 in 2019, showing a statistically significant decreasing trend of 81.86%. The highest incidence of malaria was observed during this 2000s. There are five join points in the trend of incidence from 1990 to 2019: 1995, 2000, 2006, 2010, and 2014. The standardized incidence rate showed a significant upward trend from 1995 to 2000 (APC = 27.96%, t = 19.87, *p* < 0.01), a downward trend from 2000 to 2006 (APC = −11.03%, t = −11.01, *p* < 0.001), another sharp downward trend from 2006 to 2010 (APC = −33.84%, t = −23.63, *p* < 0.01), and a gradual slowdown in the downward trend from 2010 to 2014 (APC = −10.03%, t = −6.18, *p* < 0.01), while the trends in the other two time periods (1990–1995 and 2014–2019) were not statistically significant (Figure 1).

The mortality rate of malaria also showed an increasing and then decreasing trend from 1990 to 2019 overall, decreasing from 43.11 per 100,000 in 1990 to 6.03 per 100,000 in 2019. The standardized mortality rate decreased from 81.01 per 100,000 to 6.35 per 100,000 from 2003 to 2019 (APC = −14.24%, t = −5.53, *p* < 0.01), yet the change in standardized mortality from 1990 to 2003 was not statistically significant (Figure 2).

There was an overall downward trend in the DALY rate of malaria from 1990 to 2019, with a decline of 81.51% in 2019 compared to the DALY rate of 2655.12 per 100,000 in 1990. The standardized DALY rate showed a fluctuating upward trend from 1990 to 2002 (APC = 14.09, t = 3.02, *p* < 0.01), and a significant downward trend from 2002 to 2019 (APC = −15.10, t = −8.46, *p* < 0.01) (Figure 3).

The rate of YLDs and YLLs in 1990 was 159.12 per 100,000 and 2496.01 per 100,000, respectively, decreasing to 51.01 per 100,000 and 439.85 per 100,000, respectively, in 2019, representing a reduction of 67.94% and 82.37% compared to 1990. The rates of YLDs and YLLs also changed as a proportion of DALYs, shifting from 5.99% to 10.19% and from 94.01% to 89.81% since 1990 to 2019, respectively (Figure 4).

The age-standardized rate of YLDs showed a fluctuating downward trend from 1990 to 2019, with an overall decrease of 67.94%, and a significant upward trend in the overall YLDs rate in 2000. The rate of YLDs of malaria showed a trend of increasing and then decreasing from 1990 to 2019, and a sharp increase in the YLD rate could be found from 2000 to 2005. There are three statistically significant join points in this period, with a steep increase from 1995 to 1999 (APC = 39.48%, t = 12.35, *p* < 0.01) and a sharp decrease from 2002 to 2007 (APC = −19.78%, t = −13.29, *p* < 0.01). During the period 2007 to 2010, it continued to be a significant declining trend (APC = −26.73%, t = −8.06, *p* < 0.01). In the last stage, the downward trend slowed from 2010 to 2019 (APC = −1.93%, t = −6.21, *p* < 0.01) (Figure 5).

The age-standardized YLL rate also showed a fluctuating decreasing trend from 1990 to 2019, with a more pronounced floating change in the YLL rate compared to the YLD rate, showing an overall decrease of 94.1%, due to the great reduction in the mortality of malaria. Through Joinpoint regression analysis, the trend of the YLL rate from 1990 to 2019 can be divided into two stages, one of which was from 1990 to 2002, showing an increasing trend (APC = 14.19%, t = 2.85, *p* < 0.01), while the other was from 2002 to 2019, and showed a decreasing trend (APC = −14.90%, t = −7.71, *p* < 0.01) (Figure 6).

### 3.2. The Changes in the Burden of Malaria by Gender and Age Groups

#### 3.2.1. Changes by Gender

Incidence, mortality, and DALY rates were slightly higher in males than in females for the entire period from 1990 to 2019. However, the trends in the burden of disease for males and females were mostly similar. Compared with age-standardized DALY rate and incidence, age-standardized mortality has changed relatively little over the past 30 years (Figure 7).

The rate of YLLs from malaria increased drastically around the 2000s, with males bearing more of the burden than females. Additionally, the rate of YLLs was higher for males than females in almost all years, with the difference being more apparent before 2005. In contrast, the YLD rate was higher for females than for males in every year of the study period. Meanwhile, the YLL rate was much higher than the YLD rate during the same period, and the gap between the two became smaller after 2014, when the YLL rate declined significantly (Figure 8).

#### 3.2.2. Changes by Age

From 1990 to 2019, the incidence and DALY rate among children under 5 years old were higher than for other age groups. The trend in malaria incidence among children under 5 years old was similar to that of incidence in total; its incidence decreased from 33,546.98/100,000 in 1990 to 2457.29/100,000 in 2019 (AAPC = −2.32%, t = −0.83, *p* = 0.409 > 0.05), with a small average annual change, indicating that the impact of malaria on children under 5 has been relatively stable. The mortality in children under 5 years of age climbed over time and was higher than in all other age groups. Additionally, the total number of DALYs for children under 5 years old accounted for 68.01% of the total DALYs in 2019. The DALY rate of children under 5 years old was higher than that for other age groups in each year from 1990 to 2019.

The incidence among people aged 70 years and older decreased from 3691.91/100,000 in 1990 to 3106.05/100,000 in 2019 (AAPC = −1.02%, t = −0.79, *p* > 0.05) (Table 2). The incidence in the older age group (i.e., 70 years and older) showed a significant downward trend between 1995 and 2003; however, the incidence started to rebound again from 2004 to 2010, after which it showed a slowly decreasing trend from 2010 to 2019. (Figure 9). Patients over 70 years of age had the highest mortality rate in 1990, followed by those aged 50 to 69 years. From 1990 to 2019, although mortality decreased in all age groups, only the changes in mortality trends in the 15–49 and 50–69 age groups were statistically significant (Table 3). The 50–69 age group saw the largest decline with the most significant change. The DALY rate for all age groups reached a peak in 2000. Although the rate of DALYs showed an overall decreasing trend in all age groups from 1990 to 2019, only the age groups from 15–49 years, 50–69 years, and 70 years and older showed statistically significant changes over this 30-year period (Table 4). The trends in YLD rates were extremely similar across age groups, and since YLLs accounted for the majority of DALYs, the trends in YLL rates across age groups were extremely similar to DALY rates.

## 4. Discussion

This is the first comprehensive analysis of the malaria disease burden in STP using data from GBD 2019, covering a 30-year period (1990–2019). Between 1990 and 2019, the overall malaria disease burden in STP showed a decreasing trend. However, the trends were complex, as all indicators showed an increasing trend in the early years but a gradual decline after reaching a maximum around the year 2000. The standardized incidence, standardized mortality, and standardized DALY rate decreased by an average of 4.25%, 5.57%, and 4.06% per year, respectively, as compared to 1990.

In recent years, malaria has decreased from the 4th most common cause of death among STP residents in 2009, to the 21st in 2019 [11]. The DALYs attributed to malaria decreased from 4.54% (95% CI 0.66–14.74%) of total DALYs in STP in 1990 to 1.87% (95% CI 0.7–4%) in 2019. All indicators of disease burden reached a peak in the period from 2000 to 2005; it was also evident from the Ministry of Health of STP that there were more than 40,000 cases of malaria in 2000. The report of the Ministry of Health and the CNE described that the increase in incidence during that period was strongly associated with Plasmodium’s resistance to drugs, unfavorable climate change, population resistance to indoor residual spraying, and frequent political instability [15]. Compared to the highest period for all indicators, those indicators had reduced by more than 90% in 2019. Moreover, since 2005 there has been a consistent downward trend in disease burden, which is closely related to the three rounds of national indoor residual insecticide spraying conducted from 2004 to 2007 [16]. Meanwhile, surveillance data from 1995 to 2007 show that after the introduction of indoor residual spraying, ACTs, and LLINs in 2004, malaria outpatient and hospitalization rates for both adults and children decreased by 80–90% [17].

According to the Ministry of Health report [18], compared to 2010, malaria incidence and mortality in 2015 had decreased by 47.06%, which achieved the goal of achieving a 40% reduction in incidence, but there has been essentially no change since then. Despite the downward trend from 2010 to 2019, the crude incidence rate rose from 3098.04/100,000 in 2015 to 3281.93/100,000 in 2019, and the standardized incidence rate from rose 3183.65/100,000 to 3325.66/100,000. This is not only caused by an increase in the amount of rainfall, but also has a lot to do with the low level of compliance of residents with preventive measures (e.g., house spraying and use of mosquito nets) [15]. In recent years, indoor spraying coverage has often fallen short of targets due to an increase in the number of refusers and non-responsive homes, with the national average acceptance rate for indoor spraying being 87%, 83%, and 75% in 2005, 2006, and 2007 [19], respectively, and reaching only 66.3% coverage in 2019 [20]. According to the targets set in the GTS 2016–2030 Global Malaria Technical Strategy, indicators such as incidence and mortality are to be reduced by 50% by 2020, compared to 2015 [3]. It is clear that STP has not achieved this target and has shown ups and downs in the indicators over the years. Between 2015 and 2019, five countries have seen an estimated increase in malaria incidence, with a 10% increase in STP [1], as also confirmed in the World Malaria Report 2020.

In Africa, deaths caused by malaria occur mainly in children under 5 years of age [21]. Most of the age groups showed a sharp increase in incidence followed by a significant decrease. The age group with the highest malaria disease burden in STP is children under 5 years of age. In 2019, the number of DALYs from malaria in children under 5 accounted for 68.01% of the total number of DALYs in all age groups, and their average annual number of DALYs, as a percentage of total disability years between 1990 and 2019, was 68.04%. In addition to children’s weak immunity, their lack of awareness of mosquito bites, lack of active interference with mosquitoes, and poor knowledge of malaria may also be reasons for their susceptibility. At the same time, there are no special policies for children to prevent this disease. The trends were different for two other age groups, patients aged 50–69 years and those aged 70 years and older. The age group from 50–69 years showed the least significant fluctuation in incidence, with little change in incidence from 1990 to 2006, and only showing a relatively large decrease from 2006 to 2014.

Between 2000 and 2015, elderly patients aged 70 years and above had the lowest incidence of malaria among all age groups with a significant decrease in mortality, whereas children under 5 years old became the age group with the highest mortality rate, which is probably due to a reduced risk in the elderly due to their increased immunity to clinical infection as a result of sustained exposure to the disease [22].

During the period from 1990 to 2019, all indicators, such as incidence, mortality, and DALY rate, were slightly higher in males than in females. A possible reason for this phenomenon could be the lifestyles of males, who tended to do more outdoor activities than females, combined with the fact that there were higher rates of mosquito bites outdoors than indoors. However, the difference in the incidence, mortality, and DALY rate of malaria between these two groups was not significant. The proportion of YLL was much larger than that of YLD in early years, indicating that the disease burden of malaria in STP in earlier years was mainly from the loss of life years due to early death, which might have been caused by the absence of scientific prevention and treatment methods; however, in later years, the enhanced screening and prognostic follow-up until patients recovered led to a reduction in the number of patients with severe malaria, while also greatly reducing mortality.

In STP, malaria is first diagnosed by RDT, and patients with positive RDTs have their blood collected and their samples examined microscopically to determine the type and density of Plasmodium. STP is an aid-dependent country, so its facilities are inadequate in many aspects, including human resources, and it faces multiple difficulties in the fight against malaria. According to the strategic plan called PEN 2017–2021, the integrated malaria control program includes (1) early diagnosis and appropriate and adequate treatment of cases using Artemisinin Treatment Combinations (ATCs), (2) intermittent preventive treatment for pregnant women, (3) use of mosquito nets impregnated with long-lasting insecticides and in-net spraying, (4) information, education, and communication campaigns, (5) promotion of environmental sanitation, (6) treatment of larval breeders with biocides, and (7) strengthening epidemiological and entomological surveillance. The program’s strategy is to prevent and appropriately manage malaria cases early in all health and community facilities. The results presented here also illustrate the great progress that has been made in combating malaria in the past three decades as a result of global efforts to reduce incidence and mortality of this disease [19]. This study has a few policy implications:

Firstly, international organizations such as the Global Fund, the World Health Organization, and NGOs have contributed greatly to the fight against malaria in STP. In particular, the Chinese (Guangzhou University of Traditional Chinese Medicine) Anti-Malaria Advisory Team for STP has been cooperating with the CNE to eliminate the source of malaria transmission, and thus control the spread and prevalence of malaria at the root, through universal drug administration since 2017. In 2019, the aid team and the CNE conducted a three-month MDA and long-term epidemiological testing in Bairro Liberdade, a malaria-prone village in the capital of STP, and the results showed that the parasitism rate and gametophyte carriage rate of Plasmodium falciparum in the village decreased from 28.29‰ to 0 and from 4.99‰ to 0, respectively, before the implementation of the project, which greatly suppressed malaria transmission [23]. Significant results have also been achieved in subsequent projects, which have also contributed to the reduction of the disease burden of malaria in recent years. Therefore, MDA is also considered by the Ministry of Health of STP as strong support to achieve the local goal of malaria elimination by 2025. The active-detection mission adopted by STP is important to minimize the outbreak of malaria epidemics through timely screening of the environment and population around positive cases, and this activity has been the main reason for the decline in the incidence of STP in recent years.

Secondly, it is important not only to expand the population-level coverage of malaria prevention interventions, such as insecticide-treated nets, indoor residual insecticide spraying, and simple management control, but also to implement targeted measures for different local populations in STP, especially children under 5 years old, to effectively reduce the disease burden of malaria in the population. It is necessary to strengthen the improvement of the environment, since many of the peddles that are scattered throughout villages are untreatable, and there is still a large amount of wastewater accumulation next to some residents’ houses. The pavement conditions in the villages of STP are mostly undeveloped, without completely flat surfaces, and the landscape is undulating, with low-lying areas forming pools of standing water for long periods after rain, providing conditions for mosquito breeding.

Thirdly, there is an urgent need for STP to build and improve information systems to provide more convenience and assistance for malaria prevention and control, as well as case tracking, detection, and tracing. Even though the goal of malaria elimination by 2025 appears difficult to achieve at this time, the implementation of integrated malaria control measures remains important in reducing the disease burden of malaria, and only the sustained and effective implementation of these efforts will lead to the progressive achievement of the goal. For countries with poor infrastructure, such as STP and even more African countries, when dealing with malaria and other diseases transmitted by vector, in addition to providing more scientific medical help itself, we can also address the root of the problem by improving the living environment, enhancing education, etc.

Although GBD 2019 provides important estimates of the global burden of disease, the study has key limitations in the coverage and quality of the STP database used. Due to the lack of certain stand-alone standardization of malaria case information in STP, deficiencies and inadequacies in data statistical technology and other aspects have long prevented timely statistics, timely and complete traceability, and timeliness of disease control. Most of the data in this study were estimated by model prediction, and the data sources of the model were case registries or sample surveys, so the specific values estimated by each model-based prediction will have small deviations, but the overall trend is consistent with the basic results given by the local Ministry of Health. More accurate and targeted data should be used for future analysis, with more accurate case reporting information and improved health information systems, to provide more effective research evidence for eliminating local malaria epidemics and complementing malaria control strategies [24].

## 5. Conclusions

The metrics estimated by the GBD 2019 allowed for a better understanding of the burden of malaria in STP. From 1990 to 2019, malaria incidence (1990–2000), mortality (1990–2003), and disability-adjusted life year (DALY) rates (1990–1999) in STP showed an overall increasing trend, followed by an overall decreasing trend in the subsequent period to 2019. Although indicators such as malaria incidence have declined significantly in the country, there is still a need to maintain control and surveillance measures in priority transmission areas, actively expand MDA coverage, strengthen the implementation of mosquito vector control measures, take necessary seasonal malaria prevention and control measures, and enhance the accessibility of medical services such as diagnosis and treatment.

## Figures and Tables

**Figure 1 ijerph-19-14817-f001:**
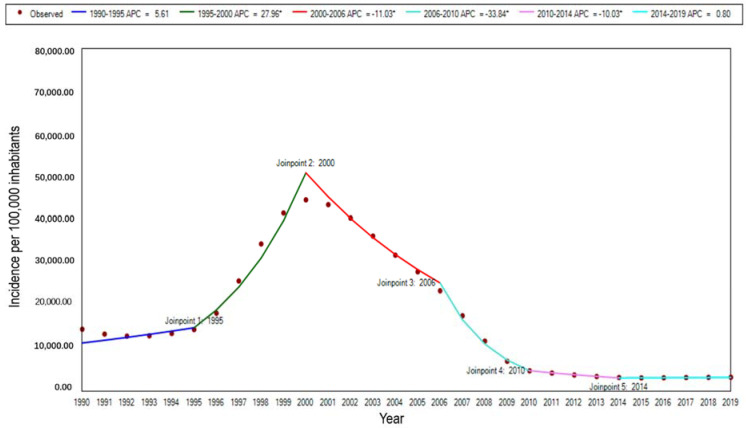
The trend of age-standardized incidence rates for malaria in STP, 1990–2019. Note: * indicates *p* < 0.05.

**Figure 2 ijerph-19-14817-f002:**
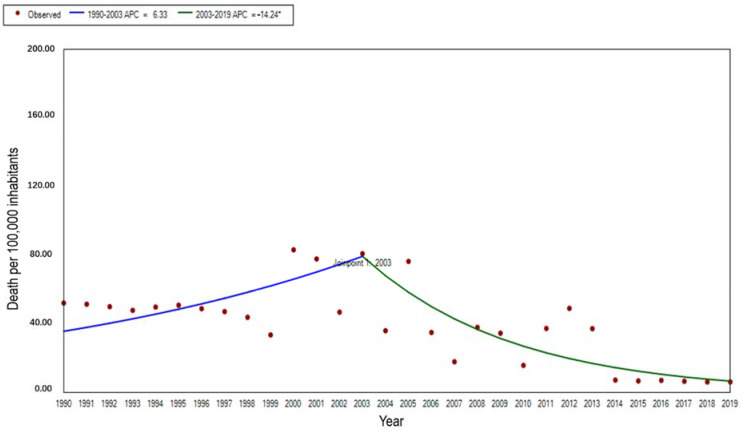
The trend of age-standardized mortality rates for malaria in STP, 1990–2019. Note: * indicates *p* < 0.05.

**Figure 3 ijerph-19-14817-f003:**
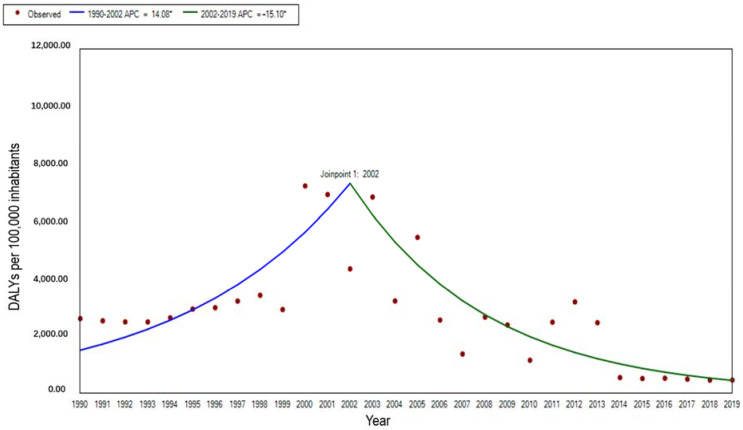
The trend of age-standardized DALY rate for malaria in STP, 1990–2019. Note: * indicates *p* < 0.05.

**Figure 4 ijerph-19-14817-f004:**
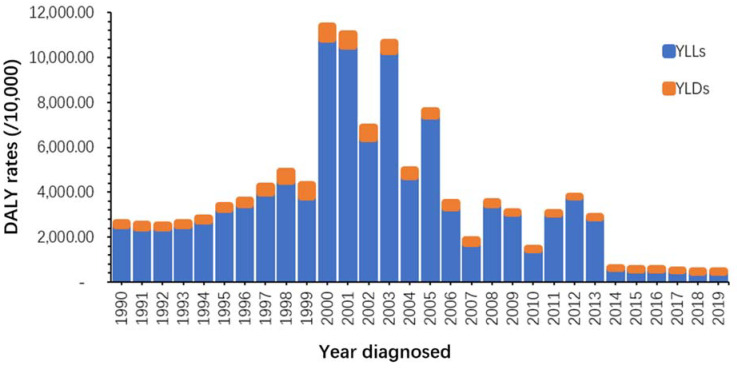
Total YLD rate and YLL rate per 100,000 (DALYs) in STP, 1990–2019.

**Figure 5 ijerph-19-14817-f005:**
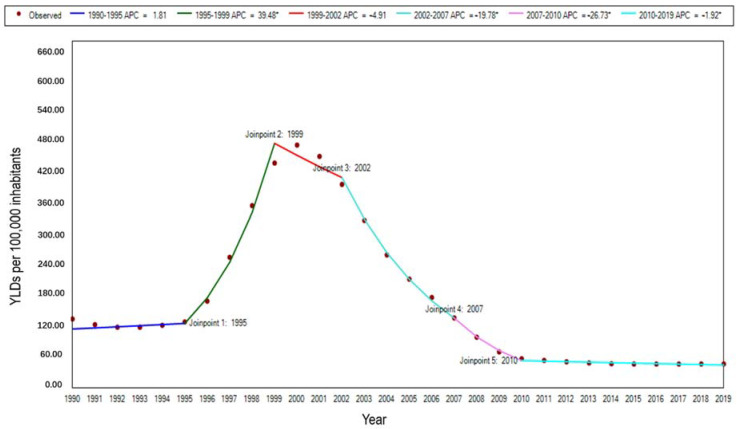
The trend of age-standardized YLD rate for malaria in STP, 1990–2019. Note: * indicates *p* < 0.05.

**Figure 6 ijerph-19-14817-f006:**
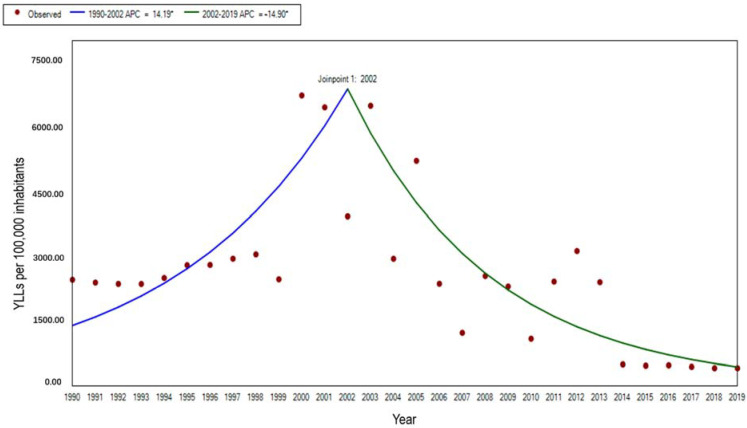
The trend of age-standardized YLL rate for malaria in STP, 1990–2019. Note: * indicates *p* < 0.05.

**Figure 7 ijerph-19-14817-f007:**
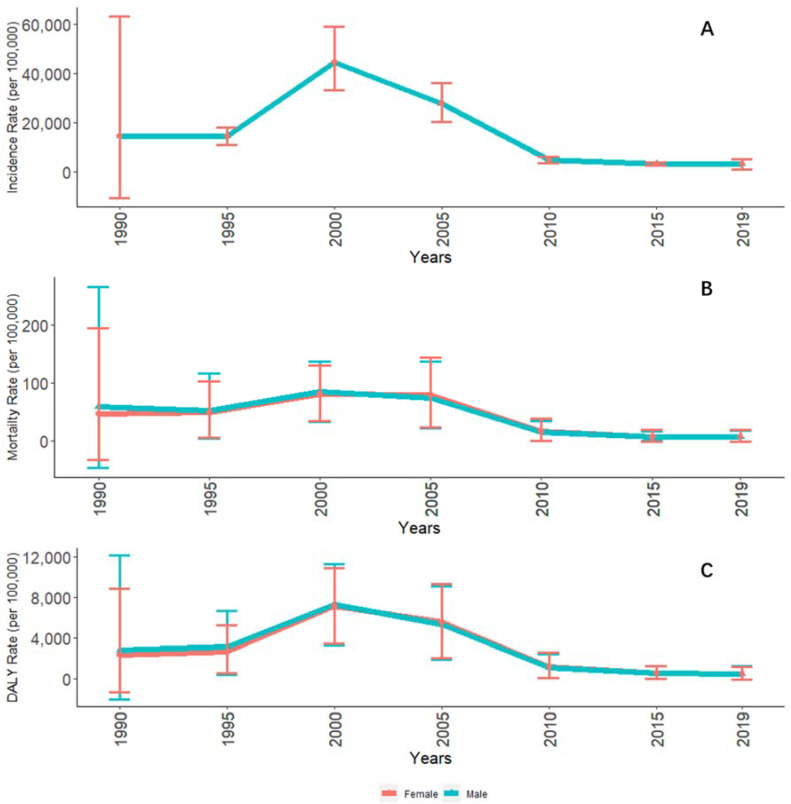
Age-standardized malaria incidence, mortality, and DALY rate in males and females in STP between 1990 and 2019. Note: (**A**) Incidence, (**B**) Mortality, (**C**) DALYs per 100,000 inhabitants from 1990 to 2019.

**Figure 8 ijerph-19-14817-f008:**
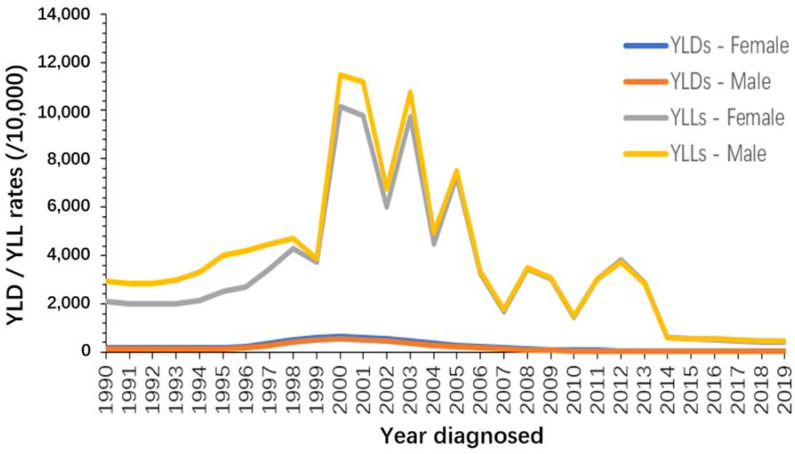
The trend of YLD rate and YLL rate of malaria, STP, 1990–2019.

**Figure 9 ijerph-19-14817-f009:**
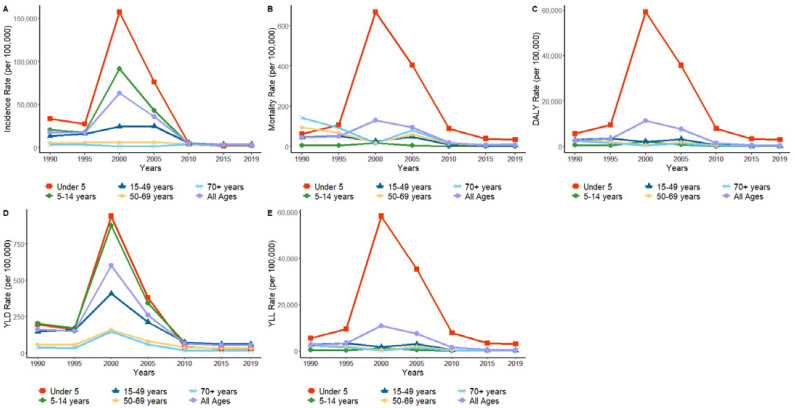
Age-standardized rates per 100,000 inhabitants from 1990 to 2019. Note: (**A**) Incidence, (**B**) Mortality, (**C**) DALYs, (**D**) YLDs, and (**E**) YLLs per 100,000 inhabitants from 1990 to 2019.

**Table 1 ijerph-19-14817-t001:** Trends of the crude rate and age-standardized rate per 100,000 for malaria among Sao Tome and Principe males and females in the period from 1990–2019.

	1990	1995	2000	2005	2010	2015	2019	1990–2019
								Relative Change (%)
CIR	18,089.54	17,248.51	63,243.40	36,205.95	4799.02	3098.04	3281.93	−81.86
CMR	43.11	49.26	128.54	95.45	18.60	7.19	6.03	−86.02
DALY Rate	2655.12	3397.66	11,394.78	7643.16	1504.96	597.26	490.85	−81.51
ASIR	14,512.31	14,438.51	44,647.26	27,896.06	4819.97	3183.65	3325.66	−77.08
ASMR	52.20	50.93	83.24	76.47	15.95	6.89	6.35	−87.84
ASDR	2601.49	2934.76	7220.89	5436.51	1152.30	518.42	462.92	−82.21

Note: CIR, crude incidence rate; CMR, crude mortality rate; ASIR, age-standardized incidence rate; ASMR, age-standardized mortality rate; and ASDR, age-standardized DALY rate.

**Table 2 ijerph-19-14817-t002:** Changes in incidence of malaria per 100,000 population by age group in STP, 1990–2019.

	1990	1995	2000	2005	2010	2015	2019	APC (%)	95% CI
**Ages**									
**<5 years**	33,546.98	27,378.15	157,419.85	76,221.54	4330.17	2262.33	2457.29	−8.4 ^a^	(−10.8, −6)
**5–14 years**	20,656.17	17,679.31	91,584.68	43,317.22	3979.95	2395.90	2536.42	−6.8 ^a^	(−8.9, −4.6)
**15–49 years**	13,595.59	16,125.09	24,514.12	24,987.67	5483.41	3662.17	3812.33	−4.2 ^a^	(−4.7, −3.7)
**50–69 years**	5665.86	5820.32	5882.16	6285.00	4472.32	3248.68	3349.68	−1.8 ^a^	(−2, −1.7)
**≥70 years**	3691.91	3140.14	1645.66	1596.39	1924.98	3020.50	3097.37	−0.6 ^a^	(−1.2, −0.1)
**Both**	18,089.54	17,248.51	63,243.40	36,205.95	4799.02	3098.04	3281.93	−5.6	(−6.9, −4.2)

Note: a indicates *p* < 0.05.

**Table 3 ijerph-19-14817-t003:** Changes in mortality of malaria per 100,000 population by age group in STP, 1990–2019.

	1990	1995	2000	2005	2010	2015	2019	APC (%)	95% CI
**Ages**									
**<5 years**	61.74	106.42	669.65	404.65	88.88	37.75	33.22	−0.9	(−4.4, 2.6)
**5–14 years**	5.10	3.62	17.45	4.93	0.40	0.14	0.12	−12.4	(−28.9, 7.9)
**15–49 years**	44.73	52.17	25.46	46.69	7.53	3.12	2.67	−8.9 ^a^	(−11.3, −6.5)
**50–69 years**	94.56	67.97	17.06	59.75	14.58	6.44	6.15	−12 ^a^	(−18.9, −4.6)
**≥70 years**	141.97	92.27	15.98	79.96	16.72	7.44	11.42	−10.8	(−22.6, 2.9)
**Both**	52.20	50.93	83.24	76.47	15.95	6.89	6.35	−6.2 ^a^	(−9.6, −2.6)

Note: a indicates *p* < 0.05.

**Table 4 ijerph-19-14817-t004:** Changes in DALY rates per 100,000 people by age group in STP, 1990–2019.

	1990	1995	2000	2005	2010	2015	2019	APC (%)	95% CI
**Ages**									
**<5 years**	5599.56	9504.15	59,194.34	35,690.86	7827.09	3340.23	2939.79	−1	(−4.5, 2.5)
**5–14 years**	608.33	458.09	2277.93	736.05	99.58	67.11	65.17	−8.1	(−15.8, 0.4)
**15–49 years**	2882.10	3429.81	2062.17	3141.75	536.33	249.84	221.46	−8.5 ^a^	(−10.5, −6.5)
**50–69 years**	2903.33	2087.28	671.10	1901.75	486.91	231.44	220.92	−11 ^a^	(−17.1, −4.5)
**≥70 years**	2346.76	1587.78	436.33	1407.74	318.93	146.98	204.38	−10.7 ^a^	(−16.8, −4.3)
**Both**	2655.12	3397.66	11,394.78	7643.16	1504.96	597.26	490.85	−5 ^a^	(−8.3, −1.6)

Note: a indicates *p* < 0.05.

## Data Availability

The GBD 2019 data are available at the GBD website (Global Burden of Disease (GBD 2019))|Institute for Health Metrics and Evaluation (healthdata.org) (accessed on 25 February 2022)).

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
