# Peer review of "Burden of Malaria in Sao Tome and Principe, 1990–2019: Findings from the Global Burden of Disease Study 2019"

_ijerph, 2022, doi:10.3390/ijerph192214817_

Round 1

Reviewer 1 Report

The present study titled “Burden of Malaria in Sao Tome and Principe, 1990-2019: Findings from the Global Burden of Disease Study 2019” deals with the progress for combating malaria in Sao Tome and Principe in the last three decades. Interesting results indicate a peak of prevalence and mortality around 2000’, an important role of host age and an important decrease of malaria incidence in the last ten years. Authors clearly explain all the process for their study and discuss results accordingly. I have very little things to say, hope they help.

Minor comments:

1 . Title too long. Too much information. Twice the same word.

2. Maybe the Abstract background is too short. The reader would like to know more about some previous information in the area and why that study should take place.

3. Lines 55-57: Include a reference after this sentence, please.

Figures 1, 2, 3, 5, 6, 7: The quality of all these images are too low but maybe is a problem of the first submission. Also, I suggest increasing letter size for a better understanding

Lines 187: Please, place the point after the parenthesis.

Line 396: It should be “More” instead of “more”.

Major comments:

In the discussion section I miss a little bit a more evolutionary explanation about the reason of why malaria parasites “prefer” younger host rather than older ones (lower prevalence in elderly population than in young ones). Maybe infected vectors are not attracted by older hosts? Or maybe because older people has been infected previously (years ago) with malaria?

Author Response

Point 1: Title too long. Too much information. Twice the same word.

Response 1: I'm sorry, I think I've simplified the topic to its most understandable state, and removing any part of it I think would not express the main idea of the article.

Point 2: Maybe the Abstract background is too short. The reader would like to know more about some previous information in the area and why that study should take place.

Response 2: Background: Malaria is a parasitic infection transmitted by mosquito vectors, commonly found in tropical regions, and characterized by high morbidity and mortality. It causes a heavy disease burden in Sao Tome and Principe (STP), which was once a high incidence of malaria, an island country in West Africa. (Lines 10-12)

Point 3: Lines 55-57: Include a reference after this sentence, please.

Response 3: Already added the reference after this sentence.

Point 4: Figures 1, 2, 3, 5, 6, 7: The quality of all these images are too low but maybe is a problem of the first submission. Also, I suggest increasing letter size for a better understanding.

Response 4: I have adjusted the size of the text title in the image, please check.

Point 5: Lines 187: Please, place the point after the parenthesis.

Response 5: Already revised this error.

Point 6: Line 396: It should be “More” instead of “more”.

Response 6: Already revised this error.

Major comments: In the discussion section I miss a little bit a more evolutionary explanation about the reason of why malaria parasites “prefer” younger host rather than older ones (lower prevalence in elderly population than in young ones). Maybe infected vectors are not attracted by older hosts? Or maybe because older people has been infected previously (years ago) with malaria?

Response: I think I explained the reason for this in lines 336-339 of the article is that older people have had malaria, and appropriately their immunity is boosted and the symptoms produced by malaria are less severe when they are infected.

Attachment: Revised Manuscript.

Reviewer 2 Report

The paper investigates malaria burden in Sao Tome and Prince between 1990 and 2019, suggesting as contributions for the literature the analysis by gender and age groups. The data source is the Global Burden of Disease 2019. I have some concerns and suggestions for the writers to clarify their results and analysis.

- Please add the information to the paper: is the access to malaria diagnosis and treatment available to everyone in the country? Is it freely accessible? This would help to understand limitations in data.

- What are the environmental characteristics that make the study area have historically a high burden of the disease? By the way, the statement that “The environment is terrible” is neither appropriate nor correct. We want more information about the most common breeding sites and the vector species in the study area.

- What are the data sources to conduct the age-standardized indicators? Does the study area produce reliable population figures or estimates? Are these figures available for which years? How about the other years (for which there was no population data or estimates)? How have the authors proceeded to fill population data gaps (if there were any)?

- The model fit presented in the Figures 2, 3 and 6 is not the best possible, since the points diverge considerably from the estimated trends. The high variability of data from one year to the other suggest issues in malaria data and/or standardization. How authors explain that? Results should be discussed carefully. There is an information about bad data quality in the end of the paper. But how bad is it? Are there estimates of how far it is from reality (in studies conducted in Sao Tome and Prince or in other countries that used the same data)? Since all the conclusions of the paper are based on this data source, it should be analyzed thoroughly by the authors.

- Figure 4: years of life lost due to premature death (YLLS) varied much more than years of life lost due to disability (YLDS). Why? Is this finding similar to other study areas or this is a singularity?

- Figure 7: why are the confidence intervals so large for 1990?

- Are children more exposed to the malaria vector? Would this explain the difference of incidence, mortality and DALY rate for 0-4 years? Where are they being infected? Why? Are there public policies in place directly to protect this age group?

- Now what? Considering the low levels of malaria incidence in the area, could we say Sao Tome and Prince reached a malaria elimination phase? What are the most appropriate diagnosis for the current stage of malaria epidemiological situation found in the study area? Is microscopy enough? Is there availability of PCR? How to move forward? We know from experience that the last steps of the malaria elimination are the hardest ones.

- What can an international audience (researchers and stakeholders) learn from the experience of Sao Tome and Prince? What surveillance and control strategies could be adapted to elsewhere? What are the limitations of replicating the successful policies implemented there in other malaria endemic countries?

Please add the information requested directly to the text, to make it explicit to the readers.

Author Response

Point 1:Please add the information to the paper: is the access to malaria diagnosis and treatment available to everyone in the country? Is it freely accessible? This would help to understand limitations in data.

Response 1:To ensure access to malaria treatment and diagnosis for the population, a Ministerial Order was issued in 2008 that mandates free ACTs, as well as free public diagnosis of malaria in the public- sector for all age groups [5]. (Already add it in the manuscript at lines 58-60)

Point 2:What are the environmental characteristics that make the study area have historically a high burden of the disease? By the way, the statement that “The environment is terrible” is neither appropriate nor correct. We want more information about the most common breeding sites and the vector species in the study area.

Response 2:The pavement conditions in the villages of Sao Tome and Principe is mostly undevel-oped, without a complete flat surface, and the landscape is undulating, with low-lying areas forming long periods of standing water after rain, providing conditions for mosquito breeding. (Lines :394-397)

Point 3: What are the data sources to conduct the age-standardized indicators? Does the study area produce reliable population figures or estimates? Are these figures available for which years? How about the other years (for which there was no population data or estimates)? How have the authors proceeded to fill population data gaps (if there were any)?

Response 3: The age-standardized data we extracted directly from the calendar year data provided by GBD, but the specific description of the annual population data in the text is based on the calendar year projections based on the 2012 National Census data from the National Institute of Statistics of São Tomé and Principe, which is also the date of the most recent census, and the annual incremental growth rates based on the calendar year projections, as detailed in the annex (published by the National Institute of Statistics).

Point 4: The model fit presented in the Figures 2, 3 and 6 is not the best possible, since the points diverge considerably from the estimated trends. The high variability of data from one year to the other suggest issues in malaria data and/or standardization. How authors explain that? Results should be discussed carefully. There is an information about bad data quality in the end of the paper. But how bad is it? Are there estimates of how far it is from reality (in studies conducted in Sao Tome and Prince or in other countries that used the same data)? Since all the conclusions of the paper are based on this data source, it should be analyzed thoroughly by the authors.

Response 4: Regarding the fit, it was the best model finally selected based on the trend analysis recommended by joinpoint software. We tried to divide into more stages for segmentation analysis, but this could not show the overall trend, and in the case of setting 5 joinpoints, the change in each stage was not statistically significant, so we chose a meaningful fitting model to represent the overall trend, but showing it in the picture would show a poor fit. The overall trend of the data extracted from GBD2019 and the data reported by the local health department are generally consistent, but the data estimated by GBD2019 are slightly higher than those presented by the local health department. For the aspect of poor data quality, due to the imperfect information reporting system and the different levels of equipment equipped in each health institution, the information reporting is incomplete and incomplete, and the information statistics are mainly conducted through the paper version of registration, so there are problems such as information omission in this process, and here we can give an example, we found that in the process of conducting household questionnaire survey based on the information of positive patients provided by the statistical department, there are Many patients with the same period of illness were not registered, which is the main reason for the poor quality of the data and the incomplete accuracy of the data. Moreover, the local health department eventually divided the age group into only two stages: under 14 years old and over 14 years old, so a detailed analysis of each age group was not possible.

Point 5: years of life lost due to premature death (YLLS) varied much more than years of life lost due to disability (YLDS). Why? Is this finding similar to other study areas or this is a singularity?

Response 5: YLLs greater than YLDs have similarities to other regions of the study area and are more common in countries with lower prevalence, as explained in the discussion section of the manuscript.

Point 6: Figure 7: why are the confidence intervals so large for 1990?

Response 6: Probably due to the small sample size in the early years, the study started late and we can also see some similarity in all the data in 1990.

Point 7: Are children more exposed to the malaria vector? Would this explain the difference of incidence, mortality and DALY rate for 0-4 years? Where are they being infected? Why? Are there public policies in place directly to protect this age group?

Response 7: In addition to children's weak immunity, the awareness of the bite and the lack of active interference with mosquitoes and the poor knowledge of malaria may also be the reasons for their susceptibility. And at the same time, there are no special policies for children to prevent this disease.

Point 8: Now what? Considering the low levels of malaria incidence in the area, could we say Sao Tome and Prince reached a malaria elimination phase? What are the most appropriate diagnosis for the current stage of malaria epidemiological situation found in the study area? Is microscopy enough? Is there availability of PCR? How to move forward? We know from experience that the last steps of the malaria elimination are the hardest ones.

Response 8: STP has been working in recent years to eradicate malaria by 2025, but at present this goal seems almost impossible to achieve. And according to a malaria case management protocol developed by the STP government (revised in 2018), the country plans to eliminate local malaria cases by 2025.

In STP, malaria is first diagnosed by RDT, and patients with positive RDTs have their blood collected and their samples examined microscopically to determine the type and density of Plasmodium. São Tomé is an aid-dependent country, so its facilities are inadequate in many aspects, including human resources, and it faces multiple dif-ficulties in the fight against malaria. (Lines 353-357 )

Point 9: What can an international audience (researchers and stakeholders) learn from the experience of Sao Tome and Prince? What surveillance and control strategies could be adapted to elsewhere? What are the limitations of replicating the successful policies implemented there in other malaria endemic countries?

Response 9: The active detective mission adopted by STP is important to minimize the outbreak of malaria epidemics through timely screening of the environment and population around positive cases, and this activity has been the main reason for the decline in the incidence of STP in recent years. (Lines 382-386)

For countries with poor infrastructure such as STP and even more African countries, in dealing with malaria and other vector infectious diseases, in addition to providing more scientific medical help itself, we can also start from the root of the problem by improving the living environment and enhancing education, etc. (lines 404-407)

Attachment: Revised Manuscript.
